# Accum™ Technology: A Novel Conjugable Primer for Onco-Immunotherapy

**DOI:** 10.3390/molecules27123807

**Published:** 2022-06-13

**Authors:** Abed El-Hakim El-Kadiry, Simon Beaudoin, Sebastien Plouffe, Moutih Rafei

**Affiliations:** 1Department of Pharmacology and Physiology, Université de Montréal, Montreal, QC H3T 1J4, Canada; abed.kadiry@gmail.com; 2Defence Therapeutics Inc., Research and Development Branch, Vancouver, BC V6C 3L6, Canada; beaudoins1983@gmail.com (S.B.); splouffe@defencetherapeutics.com (S.P.); 3Department of Microbiology, Infectious Diseases and Immunology, Université de Montréal, Montreal, QC H3T 1J4, Canada; 4Molecular Biology Program, Université de Montréal, Montreal, QC H3T 1J4, Canada

**Keywords:** Accum, cholic acid, nuclear localization signal, endosome entrapment, compromised activity, primer, functionalizing biotechnology, intracellular accumulation, immunogenicity, cancer

## Abstract

Compromised activity is a common impediment for biologics requiring endosome trafficking into target cells. In cancer cells, antibody-drug conjugates (ADCs) are trapped in endosomes or subsequently pumped extracellularly, leading to a reduction in intracellular accumulation. In subsets of dendritic cells (DCs), endosome-engulfed antigens face non-specific proteolysis and collateral damage to epitope immunogenicity before proteasomal processing and subsequent surface presentation. To bypass these shortcomings, we devised Accum™, a conjugable biotechnology harboring cholic acid (ChAc) and a nuclear localization signal (NLS) sequence for endosome escape and prompt nuclear targeting. Combined, these mechanisms culminate in enhanced intracellular accumulation and functionalization of coupled biologics. As proof-of-principle, we have biochemically characterized Accum, demonstrating its adaptability to ADCs or antigens in different cancer settings. Additionally, we have validated that endosome escape and nuclear routing are indispensable for effective intracellular accumulation and guaranteed target cell selectivity. Importantly, we have demonstrated that the unique mechanism of action of Accum translates into enhanced tumor cytotoxicity when coupled to ADCs, and durable therapeutic and prophylactic anti-cancer immunogenicity when coupled to tumor antigens. As more pre-clinical evidence accumulates, the adaptability, unique mechanism of action, and high therapeutic potency of Accum signal a promising transition into clinical investigations in the context of onco-immunotherapy.

## 1. Introduction

As medical demand and technological advancements balloon, the biotechnology market is foreseen to expand at a compound annual growth rate of 12.66%, reaching USD 106.75 billion by 2028 [1]. Oncology is the disease area that responsible for the largest share of revenue—almost half—from the development of biologics [2]. Biologics for cancer treatment chiefly include immunotherapies [3], such as monoclonal antibodies (mAbs), vaccines, and oncolytic viruses, which have transformed cancer care, providing patients with significant survival and quality of life benefits [4]. These therapeutic modalities, however, remain logistically, mechanistically, and biologically challenged, with disparate limitations hindering further clinical development and licensing [5]. Bypassing these challenges in cancer settings has thus become, and remains, the mission of multitudinous technology platforms that incorporate exosomes, chimeric antigen receptors (CARs), organoids, viruses, stem cells, cellular vaccines, antigen vaccines, mAbs, and ADCs, among others [6,7,8,9].

Accum™ is a novel biotechnology exemplifying, in its initial concept, the need to circumvent the biological challenges of ADCs [10,11,12,13,14,15]. Despite their targeted therapy nature (a receptor-specific mAb linked to a cytotoxic payload), ADCs become captives of the cellular transport mechanisms they were designed to exploit to deposit their cargo in a trojan horse-like fashion [9]. Cellular transport pathways (primarily receptor-mediated internalization followed by endosomal-lysosomal trafficking [16]) have been shown to develop resistance to ADCs, reducing their intracellular deposition and ultimately compromising their activity [9,17]. Indeed, imaging data show that only a minimal fraction of ADCs reaches its target [18,19,20]. In addition, off-target toxicities are largely seen even with FDA-approved ADCs [21,22]. This has prompted vigorous research on ADC modifications that could keep endosome entrapment in check. ADCs equipped with cell-penetrating peptides (CPPs) were effective, yet their nontarget tissue accumulation manifested a drawback [23,24,25,26,27,28]. Contrarily, ADCs equipped with pH-sensitive polymers maintained both endosome escape and target cell selectivity [29,30]. However, whether endosome escape corresponded to elevated intracellular payload accumulation—the centroid maintaining clinical significance—still begged answers. Other modification attempts consequently emerged, including ADCs equipped with peptides containing compartment-localizing amino acids, such as the NLS sequence, or segments thereof [31], that directs protein transport into the nucleus [32,33,34]. Nevertheless, entrapment in endosomes or other recycling pathways resurfaced [35,36,37].

Accum™ employs a ChAc-NLS fusion compound that enables its conjugate to override endosome entrapment and accumulate intracellularly in target cells [12,15]. The basis for exploiting ChAc, the bile acid moiety of Accum™, is that it allows nonenveloped viruses to escape endosomes [38] by inducing the metabolism of endosomal membrane sphingomyelin into ceramide, appending subsequent structural and dynamic membrane changes that promote endosome-to-cytoplasm protein traversal without killing cells [39,40,41,42]. On the other hand, the NLS moiety of Accum™ redirects trafficking of the conjugate to the nucleus; ultimately, both Accum moieties increase intracellular retention in target cells [12]. The concept of the ChAc-NLS fusion compound serving as a cell accumulator (Accum for short) has been shown to be pliable, with the potential for coupling to different molecular conjugates for use in different cancer settings [10,11,12,13,14,15]. In this review article, we discuss the genesis of Accum™ and highlight its docility in producing context-specific therapeutic strategies. Furthermore, we recapitulate the data reported hitherto on the validity and efficacy of Accum™ as a functionalizing technology that elevates the therapeutic potency of various molecular conjugates. Finally, we underline the advantages of Accum over other conjugable molecules in the framework of previous literature reports.

## 2. Accum™: Genesis of a Primer

Assimilating their previous studies [31,35,36,37,43], wherein a synthetic peptide harboring the classical NLS (cNLS) sequence amply dictated the nuclear translocation of a mAb conjugate, Beaudoin and colleagues [12] coupled ChAc to the non-CPP, CGYGPKKKRKVGG [44], generated by an automated peptide synthesizer, then processed and characterized the conjugate via mass spectroscopy and ultra-performance liquid chromatography. PKKKRKV represents the cNLS sequence taken from simian virus 40 (SV-40) large T antigen, while GYG and GG residues serve as N- and C-terminus spacers, respectively [12]. Conjugation of Accum to protein can be performed by using any conjugation technique, such as the cross-linker sulfosuccinimidyl-4-(N-maleimidomethyl)-cyclohexane-1-carboxylate (sulfo-SMCC) reacting with the amine of lysine and the sulfhydryl moiety of cysteine via its NHS-ester and the maleimide reactive group of the cross-linker (Figure 1) [12,14,15]. ChAcNLS, or Accum, is water-soluble; has a positive net charge of +5 and a molecular weight of 1.8 kDa; and is synthetically recovered with ≥94% purity and a molecular mass of 1768.5 g/mol [12,14]. Accum is added to conjugates in the same, or higher, molar excess ratio to yield Accum-modified conjugates [15]. During the modification process, free cross-linker and free Accum are eliminated by multiple centricon filtration and a Sephadex column, while Accum-modified conjugates are subsequently processed in ultrafiltration tubes, concentrated, and biochemically characterized with SDS-PAGE, turbidity assays, differential scanning fluorimetry, and protein concentration assays, followed by functional evaluation with conjugate-receptor affinity assays against naked or non-Accum-modified conjugates [14]. The number of Accum per conjugate is proportional to the number of accessible lysine residues [15] and the intracellular accumulation of conjugates [12,14]. The rate of conjugation is estimated by SDS-PAGE gel [12,14].

Accum can be likened to a polymath primer: it is well-informed about the biological limitations of CPP- and NLS-equipped conjugates [10] and highly adaptable for functionalizing different conjugates for different therapeutic purposes [10,11,12,13,14,15]. For instance, Beaudoin et al. [12] designed 7G3-Accum—Accum coupled to 7G3—a mAb specifically targeting interleukin-3 receptor-α (IL-3Rα), a cell-surface antigen expressed by leukemic cells. In the context of muscle invasive bladder cancer (MIBC), the same team [13,14] designed A14-Accum, where A14 is a mAb specific for interleukin-5 receptor α-subunit (IL-5Rα). In another therapeutic context, Accum was coupled to the clinically approved ADC trastuzumab-emtansine (T-DM1) [10]. Recently, Bikorimana et al. [15] introduced Accum to the setting of cellular vaccines, coupling ChAcNLS to the xenoantigen ovalbumin (OVA) with the rationale of triggering potent antigen presentation by DCs. The same group further linked Accum to lymphoma tumor lysate proteins to instill potent anti-tumor immunity in DCs. Overall, these studies attest to the pliability of Accum as a biotechnology that primes the function of various molecular conjugates, thereby enhancing therapeutic outcomes.

## 3. What Has Accum™ Demonstrated So Far? Characterization, Validation, and Efficacy Data

Most ADCs undergo receptor-mediated intracellular transport to lysosomes, where they exploit pH-sensitive proteases for the catabolism, release, and intracellular diffusion of their cytotoxic payloads [9]. However, this transport system faces a backlash from cancer cell mechanisms, including receptor downregulation/recycling and increased expression of multidrug resistance pumps, that either entrap drug payloads in endosomes or counteract their intracellular accumulation [45,46]. On another note, several new ADC modifications, including ADCs coupled to synthetic CPPs or pH-sensitive polymers, have been effective in evading endosome entrapment or routing to alternative cellular compartments; however, they have failed to either produce intracellular retention [29,30,47] or achieve target cell specificity and sufficient tumor uptake [23,24,25,26,27,28]; otherwise, they have induced high toxicity by indiscriminately penetrating off-target cells [24,47]. The Accum fusion compound ChAcNLS was thus devised with the aim of enabling ADCs to (i) dodge entrapment in the endosomal-lysosomal pathway, (ii) partake in nuclear routing, and (iii) accumulate intracellularly while retaining target cell selectivity and evading off-target payload effects [12]. Optimum endosome escape and intracellular accumulation was hypothesized to be the result of synergism between (a) the ChAc moiety, which selectively disrupts endosomal membranes by triggering ceramide formation while leaving plasma membranes intact [12] in the same way it allows nonenveloped viruses to escape from endosomes into the cytoplasm [38,39,40,41,42]; and (b) the NLS moiety, which directs molecular conjugates toward the nucleus [12]. To validate these hypotheses, different in vitro and in vivo studies were performed using different Accum-modified conjugates. Additional studies were also conducted to evaluate whether the distinctive cellular delivery mechanism of Accum provides superior therapeutic advantages compared to control counterparts.

### 3.1. 7G3-Accum

Beaudoin et al. [12,13] initially designed 7G3-Accum, an Accum-modified mAb specific for IL-3Rα. In characterization assays, the number of Accum copies per antibody was shown to correspond with the sulfo-SMCC:antibody ratio, reaching a 50:1 ratio with an average of 8.5 and 2.6 copies per heavy and light chain of 7G3, respectively [12,13,15]. In proof-of-concept experiments, IL-3Rα-expressing TF-1a leukemic cells were used to study the intracellular trafficking of 7G3-ChAcNLS against different controls, including unmodified 7G3, 7G3-ChAc, 7G3-NLS, 7G3-ChAcLeu, and 7G3-ChAcDap (ChAcLeu and ChAcDap enable endosome escape—but with less efficacity—without nuclear trafficking, and possess, respectively, structural and net charge similarities to ChAcNLS). Indeed, confocal microscopy revealed a distinctive cellular distribution pattern for 7G3-ChAcNLS, which was centered and homogenously radiating throughout the target cell, suggesting intracellular antibody retention [12,13]. Contrarily, 7G3- and 7G3-NLS-treated cells both showed similar distribution patterns with clusters proximal to the intracellular surface of the plasma membrane, typical of endosome entrapment [12]. On the other hand, 7G3-ChAc was detected in the cytoplasm with a homogeneous distribution throughout, but not in the nucleus. In quantitative analyses, flow cytometry showed significantly higher intracellular accumulation of 7G3-ChAcNLS compared to all controls [12]. Intracellular accumulation levels compared to unmodified 7G3 were significantly higher for 7G3-ChAcNLS (3-fold) and 7G3-ChAc (1.8-fold), whereas no significant change was detected for 7G3-NLS. The higher intracellular accumulation of 7G3-ChAcNLS compared to 7G3-ChAc, 7G3-ChAcLeu, and 7G3-ChAcDap revealed that the combination of the ChAc moiety with the NLS moiety is responsible for the endosome escape and intracellular accumulation of ChAcNLS [12]. In fact, if ChAc had only endosome escape activity and NLS only nuclear trafficking activity, the level of intracellular accumulation should have been the same as 7G3-ChAc, 7G3-ChAcLeu, or 7G3-ChAcDap [12]. To further evaluate target cell specificity, intracellular and nuclear accumulation of 7G3-ChAcNLS versus unmodified 7G3 were imaged with confocal microscopy in IL-3Rα-positive and -negative cells. The data revealed that ChAcNLS modification resulted in ample specific accumulation of 7G3 in target cells and nuclei, with a margin of 10–13% nonspecific localization. Similarly, the authors used flow cytometry, western blot, and radioactivity assays to demonstrate that Accum could enhance the intracellular and nuclear accumulation of its conjugate by a margin of 21.1-fold [12,13]. These data showed that the complementary functions of the Accum moieties—ChAc and NLS—are equally indispensable for maintaining effective intracellular conjugate accumulation and target cell selectivity.

### 3.2. A14-Accum

Paquette et al. [11] have previously shown that the rapid internalization of IL-5Rα, whose levels are preferentially elevated in MIBC tumors, renders the use of ADCs with specificity for IL-5Rα strategic for the intracellular accumulation of payloads, including the positron emitter copper-64 (^64^Cu) used for imaging. Therefrom, the same group [14] designed A14-Accum, an Accum-modified mAb against IL-5Rα, to characterize its use as a tumor cell accumulator. To further assess its use as an efficient drug delivery system, A14-Accum was coupled to the radioactive diagnostic agent ^64^Cu [13]. ^64^Cu-A14-Accum indeed revealed a purity of preparation within the 95–99% range and had an affinity for IL-5Rα in the nanomolar spectrum [14]. Unpublished data also revealed that Accum augmented the kinetics of receptor internalization. When incubated with MIBC cell lines and assessed with flow cytometry (for quantifying ceramide levels) and confocal microscopy (for visualizing endosome disruption and quantifying nuclear localization), A14-ChAcNLS demonstrated marked endosome escape with nuclear routing compared to A14 [14]. In radioactivity assays investigating cargo deposition, A14-ChAcNLS significantly increased the intracellular accumulation of ^64^Cu cargo by 3.3–9.4- and 3.2–4.6-fold compared to A14 and A14-NLS, respectively, in high- and low-density IL-5Rα-expressing MIBC cell lines [14]. In the same cell lines, A14-ChAcNLS also increased the nuclear accumulation of ^64^Cu cargo by 1.7–5.3- and 2.5–2.6-fold compared to A14 and A14-NLS, respectively. When target selectivity was analyzed against an IgG-ChAcNLS control showed, A14-ChAcNLS showed only trace non-specificity for intracellular ^64^Cu cargo accumulation [14]. Beaudoin et al. [13] also showed that Accum resulted in significant intracellular and nuclear specificity and accumulation (≥3-fold) of ^64^Cu in IL-5Rα-positive invasive bladder cancer cells. These in vitro data demonstrated that both Accum moieties are necessary for increasing the intracellular accumulation of cargo-bearing conjugates and maintaining target specificity.

Paquette and colleagues [14] further characterized the pharmacokinetic (PK) profile of ^64^Cu-A14-ChAcNLS injected into mouse models of human high- or low-density IL-5Rα MIBC. Blood sampling revealed an estimated half-life (t_1/2_) of 54.4 h, and a study of the organs indicated major hepatic metabolism and clearance with early transient glomerular accumulation. In comparative analyses, the t_1/2_ of ^64^Cu-A14-ChAcNLS was 45% lower compared to ^64^Cu-A14 (t_1/2_ = 99.5 h), indicating increased blood clearance. Nonetheless, ChAcNLS maintained the levels of ^64^Cu-A14 tumor uptake. Moreover, organ dissection 96 h post-injection showed that the biodistribution of ^64^Cu-A14-ChAcNLS was reduced in most healthy organs compared to ^64^Cu-A14 with significant reductions in the liver and heart [14]. Biodistribution of ^64^Cu-A14-ChAcNLS was also visualized with positron emission tomography (PET) and revealed equivalent tumor uptake with a higher tumor/adjacent muscle uptake ratio compared to ^64^Cu-A14 [13,14]. These preclinical PK and PET analyses demonstrate that Accum technology provides ACs with (i) comparable high rates of tumor uptake, (ii) elevated intracellular accumulation, and (iii) intermediate clearance rates from healthy tissues, for an overall improvement in selective tumor targeting.

### 3.3. T-DM1-Accum

T-DM1 clinically hinders breast cancer growth, yet several patients still experience disease progression [48] due to insufficient intracellular accumulation [17,49]. Lacasse et al. [10] thus investigated whether Accum-modification enhanced intracellular accumulation of T-DM1 by using the human epidermal growth factor receptor 2 (HER2)-positive breast cancer cell line SKBR3. Confocal microscopy with temporal imaging revealed that T-DM1-Accum first undergoes endosome entrapment, then diffuses throughout the cytoplasm before slowly localizing in the nucleus, where it is proteolyzed and T-DM1 is released [10]. In subsequent analyses, nuclear transport of T-DM1-Accum was shown to be mediated by the nuclear transport receptor importin-7 (IPO7) based on electrostatic interactions fueled by the cationic charge build-up on NLS moieties [10]. Contrarily, unmodified T-DM1 localized near the plasma membrane [10] due to the rapid recycling of HER2 back to the cell surface of SKBR3 [17,50,51]. Additionally, T-DM1-Accum had stronger cytotoxic potency compared to T-DM1-NLS and unmodified T-DM1 by several-fold. Importantly, the authors showed that the specificity of T-DM1 for HER2 was not altered by Accum modification, and Accum did not perturb HER2 binding and its internalization processes [10]. These data validated that Accum modifications prime the cytotoxicity of molecular payloads by overriding endosome entrapment and directing nuclear localization, all while retaining target cell specificity. Next, the same group [10] investigated the therapeutic potency of T-DM1-Accum in HER2-positive SKBR3 cells. T-DM1-Accum demonstrated marked cytotoxicity, achieving 50% tumor growth inhibition at concentrations >9- and >3.2-fold lower compared to T-DM1 and T-DM1-NLS, respectively [10]. In gene knockdown studies, this cytotoxicity relied on IPO7-mediated nuclear transport, likely driven by electrostatic interactions between the NLS moiety of Accum (positive charge) and IPO7 nuclear receptor (negative charge) [10]. Notably, T-DM1-NLS demonstrated >1.8-fold more cytotoxicity compared to T-DM1, and Accum was not cytotoxic itself, but rather enhanced the cytotoxicity of T-DM1, since Trastuzumab (Tmab)-Accum was as equally non-toxic as Tmab. Overall, these data corroborate that both Accum moieties operate with a functional synergism that results in enhanced cellular accumulation and therapeutic efficacy [10].

### 3.4. OVA- and Lymphoma Lysate Proteins-Accum

Antigen presenting cells such as DCs engulf soluble antigens and sort them into endosomes for limited degradation before exportation to the cytosol [52]. Therein, the proteasome further processes antigen fragments into short amino acid sequences that are presented on the cell surface by Major Histocompatibility Complex (MHC) class I molecules for T-cell activation, a process termed cross-presentation [53,54]. However, non-specific endosomal degradation of antigens in certain subsets of DCs, such as monocyte-derived CD8^−^ DCs, can compromise antigen immunogenicity and reduce anti-tumoral immunity [55]. Drawing on Accum’s innovative mechanism of action validated with 7G3 [12], A14 [14], and T-DM1 [10], Bikorimana and colleagues [15] conjugated Accum to soluble antigens to investigate whether its ADC-functionalizing biotechnology could prime the antigen cross-presentation properties of CD8^−^ DCs. Accum modification increased the resistance of OVA to thermal denaturation [15]. Unlike OVA, OVA-Accum uptake by DCs induced endosome rupture, as revealed by confocal microscopy. Using primary bone-marrow derived DCs, it was further shown that OVA-Accum was processed in the cytosol, as per flow cytometry analysis [15]. Complementary antigen presentation assays using primary DCs co-cultured with T cells also showed that OVA-Accum induced marked immune cell activation, resulting in heightened inflammatory cytokine secretion [15]. Similarly, Accum-modified proteins derived from lymphoma lysate significantly enhanced the immunogenicity of allogeneic DCs in tumor-bearing mice [15]. Combined, these data validate the peculiarity of endosome-to-cytosol translocation, where Accum enables antigens to escape endosomal damage, undergo efficient proteasomal processing into immunogenic peptides, and arrive at the cell surface for potent antigen cross-presentation and subsequent T-cell activation.

Deploying these proof-of-principle data, Bikorimana and colleagues investigated whether the enhanced immunogenicity of OVA-Accum in DCs translates to a potent cellular vaccine [15]. In the context of prophylactic vaccination, two doses of OVA-Accum-pulsed primary DCs were administered subcutaneously 2 weeks apart in mice challenged thereafter with three ascending doses of an OVA-expressing T-cell lymphoma. Compared to OVA-pulsed DCs, OVA-Accum-pulsed DCs completely counteracted tumor growth, maintaining full and durable survival of all mice at 96 days post-immunization. Concomitantly, serum analysis showed higher antibody titers, and immunophenotyping revealed markedly higher levels of CD4 effector and CD8 central and effector memory T cells. Pro-inflammatory mediators were also more heightened [15]. As for therapeutic vaccination, lymphoma-bearing mice were administered two subcutaneous injections of OVA-Accum-pulsed syngeneic or allogeneic DCs one week apart. Compared to OVA-pulsed DCs, OVA-Accum-pulsed DCs delayed tumor growth and prolonged survival. Co-administration of the immune-checkpoint inhibitor anti-PD-1 over 2 weeks further enhanced these outcomes, producing at 54 days post-immunization 50% overall survival (OS), 30–40% partial response, and 10–20% complete response (CR) [15]. Allogeneic DCs pulsed with Accum-modified non-OVA-expressing lymphoma lysate proteins were also examined as a broader-spectrum vaccine for therapeutic efficacy. In lymphoma-bearing mice, there was only a minor delay in tumor growth. However, combined with anti-PD-1, this treatment resulted in marked tumor growth delays resulting in 70% OS and 30% CR compared to standard lysate-pulsed DCs co-administered with anti-PD-1 [15]. With Accum-lysate-pulsed DCs/anti-PD-1, immunophenotyping also showed elevated levels of immune effector cells and diminished levels of regulatory CD4 T cells [15].

Interestingly, the authors went beyond cellular vaccination to assess Accum for its benefits in protein vaccination for prophylaxis. Indeed, two doses of OVA-Accum were administered subcutaneously 2 weeks apart in mice challenged thereafter with an OVA-expressing T-cell lymphoma. The authors found that OVA-Accum injections significantly delayed tumor growth, maintaining 30% animal survival beyond 1 month post-immunization [15]. The addition of adjuvants to the OVA-Accum formulation further elevated anti-tumor immunity, producing up to 100% survival at 42 days post-immunization. Concurrently, serum analysis showed significantly higher anti-OVA IgG antibody titers [15].

Overall, these animal experiments showed that Accum-modification of antigens generates prophylactically and therapeutically potent cellular and protein antigen vaccines for cancer treatment.

## 4. Discussion

Accum presents advantages over other conjugable primers, including NLS, TPP-9476, CPPs, and targeting peptides.

When coupled to 7G3-NLS, the Auger electron-emitting indium-111 (^111^In) did not induce marked cytotoxicity compared to ^111^In-7G3 that was likely due to meager nuclear localization [35,36,37,56]. Contrarily, ChAcNLS increased intracellular accumulation of 7G3 by factors of 4.1–21.1 due to the combined properties of endosome escape and nuclear localization with a matching increase in intracellularly deposited ^64^Cu radioactivity [12]. A report has previously shown that the internalization of TPP-9476, an ADC targeting IL-3Rα, was 3.5-fold higher compared to isotype mAb [57]. Our data show that internalization is 8.6-fold higher for 7G3-Accum compared to isotype mAb and up to 9.4-fold higher for A14-Accum compared to A14 [14].

Additionally, the peak of accumulation for ADCs functionalized with Accum [12] has either exceeded [58] or shown to be comparable to [25] that for ADCs functionalized with the trans-activator of transcription (TAT) sequence [58] and the membrane transport sequence (MTS) [25], respectively. TAT and MTS are both examples of CPPs, short sequences of natural or synthetic origin that have been preclinically evaluated for cancer diagnosis and therapy due to their ability to translocate cargo complexes across the plasma membrane via direct translocation or endocytosis [59,60,61]. Despite being numerous and showing promise for delivering different types of drugs, CPPs have yet to attain regulatory approvals due, in part, to pharmacokinetic handicaps hindering translation into clinical settings [62]. For instance, TAT-coupled ADCs have failed to achieve adequate tumor uptake in vivo due to rapid washout and sequestration in the spleen and liver preventing sufficient biodistribution [27,28,47]. Other CPPs have also been degraded in circulation before reaching their target [63]. With Accum on the other hand, ^64^Cu-A14 has shown intermediate clearance from blood and healthy tissues with good tumor uptake [13,14]. Other clinical limitations for CPPs, which appear to be forestalled by Accum, include non-target tissue uptake and subsequent systemic toxicity [23,24,25,26,27,28,64,65], low specificity [62], and immunogenicity [66]. It has also been shown that CPPs undergo endosomal entrapment after translocating into the cytosol [67,68], which ultimately compromises the function of coupled drugs due to inevitable targeting to either the lysosomes for degradation or the plasma membrane for recycling and exportation [69,70]; therefore, effective endosome escape and homing into specific cellular organelles have since been sought to achieve therapeutic efficacy [71,72,73].

Besides CPPs, other targeting peptides, such as p28, TCP-1, and LyP-1, rely on receptor recognition to deliver therapeutic payloads [74]. p28 is an amino acid that enables the copper-containing redox protein azurin to target cancer cells, eventually inducing cell cycle arrest [75]. In tumor-bearing subjects, it has been shown that p28 induces 6.6% CR, 20% PR, and 20% OS [76]. In tumor-bearing mice, we have shown that an Accum-OVA vaccine maintains 30% OS alone and 100% OS in combination with adjuvants. Combined with anti-PD-1, Accum-DC vaccine also induced 10–20% CR, 30–40% PR, and 50% OS [15]. TCP-1, a peptide targeting the vasculature of gastric cancer cells, was conjugated to tumor necrosis factor α (TNFα) to deliver the anticancer agent 5-fluorouracil (5-FU). The results showed that TCP-1 modification did not enhance the bioactivity of TNFα because no differences in 5-FU-induced cancer cell death were detected [77]. Tumor volume also did not significantly change [77]. Accum-modification, however, was reported to enhance the bioactivity of T-DM1, markedly increasing its cytotoxic potency [10]. The tumor homing peptide LyP-1 was conjugated to abraxane, a clinically approved paclitaxel-albumin nanoparticle, then administered to mice bearing human cancer xenografts. The data showed statistically significant results with a 1.38-fold increase in cytotoxicity [78]. With Accum, however, the cytotoxicity of the clinically approved T-DM1 was enhanced by several fold [10].

Furthermore, NLS-modified ADCs, such the epidermal growth factor receptor-positive tumor-targeting ^111^In-nimotuzumab-NLS, are infamous for their rapid plasma clearance and increased sequestration in normal tissues, which translate to reduced tumor accumulation [31,79]. However, Accum demonstrates that the combination of ChAc with NLS delays the sequestration of ADCs in healthy tissues, thereby permitting targeted and good tumor uptake relative to unmodified ADCs [13,14]. Our data on T-DM1-Accum further provide strong evidence that Accum can prime the function of tumor-targeting ADCs by evading the resistance mechanisms that affect Tmab, T-DM1, and other biopharmaceuticals [49,80,81,82]. On another note, given that ChAc has been previously conjugated to liver-targeting drugs for enhanced delivery [83,84], ChAcNLS could similarly be exploited in targeted cancer therapy to decrease cellular exportation of drugs and promote intracellular retention. Other applications for ChAcNLS-modified ADCs could include phenotype-specific tumor imaging and chemotherapeutic drug transport to tumors to achieve more specific uptake and higher therapeutic efficacy.

With Accum-modified antigens [15], endosome-to-cytosol translocation was suggested to be crucial for antigen cross-presentation, a property unattained by certain tumor associated antigen-based DC vaccines due to endosome entrapment that failed to elicit sufficient T-cell responses [85,86]. Our in vivo experiments with Accum-based vaccines also convey the potential for therapeutic application of i) Accum-linked antigens with add-on checkpoint blockers and ii) Accum-linked tumor lysate proteins, which permit the development of personalized cancer vaccines without requiring specific epitope identification.

Per scientific integrity, we stand aware of the small yet growing number of published reports investigating Accum as a conjugable primer [10,11,12,13,14,15] and the conclusion biases that might arise thereof. However, the uniqueness of this biotechnology merits early communication to the research community as it continues to unfold with further pre-clinical studies underway.

## 5. Conclusions

Accum is a novel biotechnology harnessing the properties of two moieties for enhanced intracellular accumulation. By promoting endosome escape and nuclear routing, Accum has been demonstrated to make for an effective conjugation moiety that functionalizes different molecular conjugates. When coupled to ADCs, Accum maintains target cell selectivity, shows high tumor uptake rate with intermediate clearance rates from healthy tissues, and enhances tumor cytotoxicity while minimizing off-target effects. When coupled to antigens, Accum triggers durable immunogenicity in mice and has proven strategic for creating prophylactic and therapeutic anti-cancer cellular and protein vaccines. As its properties continue to emerge with further studies for planification and execution phases, Accum is a promising candidate for clinical investigations in onco-immunotherapy settings.

## 6. Patents

The Accum™ technology is protected by 10 different patents (US 63/127,731, US 63/202,047, PCT/CA2021/051543, US 17/516,161, US 63/201,620, US 63/260,648, US 63/264,126, US 63/265,125, US 16/085,141 and US 63/256,726).

## Figures and Tables

**Figure 1 molecules-27-03807-f001:**
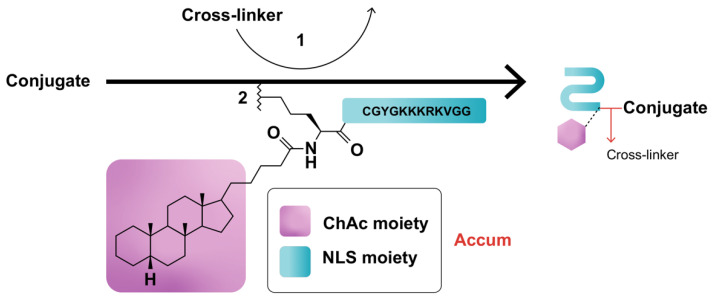
Schematic representation of Accum-modification of protein conjugates. In step (1), the protein reacts with a cross-linker, yielding a maleimide-activated conjugate. In step (2), the sulfhydryl group of the N-terminus cysteine cap of the NLS moiety of Accum reacts with the maleimide-activated conjugate, yielding an Accum-conjugate.

## Data Availability

Not applicable.

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
