# Peer review of "Accum™ Technology: A Novel Conjugable Primer for Onco-Immunotherapy"

_molecules, 2022, doi:10.3390/molecules27123807_

Round 1

Reviewer 1 Report

The study by Abed El-Hakim El-Kadiry et al. reviews Accum™ Technology: A Novel Conjugable Primer for Onco-Immunotherapy. Overall, the content of this work is interesting, There are some issues which should be addressed:

Drugs for Onco-Immunotherapy include not only antibody drugs, but also some targeting peptide drugs. What is the current status of the Conjugable targeting peptide drugs and Accum system? Author also needs to summarize and discuss this issue.

Author Response

Reviewer's Comment 1. Drugs for Onco-Immunotherapy include not only antibody drugs, but also some targeting peptide drugs. What is the current status of the Conjugable targeting peptide drugs and Accum system? Author also needs to summarize and discuss this issue.

Authors’ response 1.

Thank you for taking the time to read our manuscript and suggesting adding valuable ideas to the discussion.

Per your comment, we have added and highlighted a part pertaining to conjugable targeting peptides (“Discussion” section, lines 350-367); however, since these peptides are numerous, we limited our comparative discussion to 3 peptide examples (p28, TCP-1, LyP-1), as they allow the comparison with Accum in certain efficacy aspects. The added part reads as follows:

Besides CPPs, other targeting peptides, such as p28, TCP-1, and LyP-1, rely on receptor recognition to deliver therapeutic payloads [72]. p28 is an amino acid that enables the copper-containing redox protein azurin to target cancer cells, eventually inducing cell cycle arrest [73]. In tumor-bearing subjects, it has been shown that p28 induces 6.6% CR, 20% PR, and 20% OS [74]. In tumor-bearing mice, we have shown that Accum-OVA vaccine maintains 30% OS alone and 100% OS in combination with adjuvants. Combined with anti-PD-1, Accum-DC vaccine also induced 10%–20% CR, 30%–40% PR, and 50% OS [15]. TCP-1, a peptide targeting the vasculature of gastric cancer cells, was conjugated to tumor necrosis factor α (TNFα) to deliver the anticancer agent 5-fluorouracil (5-FU). The results have shown that TCP-1 modification did not enhance the bioactivity of TNFα, as no differences in 5-FU-induced cancer cell death were detected [75]. Tumor volume also did not significantly change [75]. Accum-modification, however, was reported to enhance the bioactivity of T-DM1, markedly increasing its cytotoxic potency [10]. The tumor homing peptide LyP-1 was conjugated to abraxane, a clinically approved paclitaxel-albumin nanoparticle, then administered to mice bearing human cancer xenografts. Data have shown statistically significant results with 1.38-fold increase in cytotoxicity [76]. With Accum, however, the cytotoxicity of the clinically approved T-DM1 was enhanced by several folds [10].    

Reviewer 2 Report

Author well characterized the features of Accum showing basic property of Accum and advanced studies including in vitro and clinical studies collaborated with immunotherapy. The benefits of AccumTM compared with other ADCs are described in only Introduction section. Detailed superior points of individual Accum-modified products such as 7G3-Accum compared with other ADCs would provide a deep understanding on AccumTM.

Author Response

Reviewer's Comment 1. Author well characterized the features of Accum showing basic property of Accum and advanced studies including in vitro and clinical studies collaborated with immunotherapy. The benefits of AccumTM compared with other ADCs are described in only Introduction section. Detailed superior points of individual Accum-modified products such as 7G3-Accum compared with other ADCs would provide a deep understanding on AccumTM.

Author’s response 1.

Thank you for your time and dedication in reading our manuscript and providing insight.

The benefits of Accum versus other ADCs are described in the Introduction as well as in the Discussion section, specifically Lines 318-389, where we compare 1) 7G3-Accum to the ADC 7G3-NLS (Lines 320-324: "Coupled to 7G3-NLS, the Auger electron-emitting indium-111 (111In) did not induce marked cytotoxicity compared to 111In-7G3 likely due to meager nuclear localization [35-37, 54]. Contrarily, ChAcNLS increased the intracellular accumulation of 7G3 by factors of 4.1–21.1 due to the combined properties of endosome escape and nuclear localization with a matching increase in intracellularly deposited 64Cu radioactivity [12]."); 2) A14-Accum to the ADC nimotuzumab-NLS (Lines 368-372: "Furthermore, NLS-modified ADCs including the epidermal growth factor receptor-positive tumor-targeting 111In-nimotuzumab-NLS are infamous for their rapid plasma clearance and increased sequestration in normal tissues, which translate to reduced tumor accumulation [31, 77]. However, Accum pinpoints that combining ChAc with NLS delays the sequestration of ADCs in healthy tissues, thereby permitting targeted and good tumor uptake relative to unmodified ADCs [13, 14]."); and 3) T-DM1-Accum to the ADC T-DM1 (Lines 373-375: "Our data on T-DM1-Accum further provide strong evidence that Accum could prime the function of tumor-targeting ADCs by evading resistance mechanisms faced by Tmab, T-DM1, and other biopharmaceuticals [49, 78-80]."). To accommodate your suggestion and and provide further understanding, we have also added and highlighted more data that compare 7G3-Accum and A14-Accum to the ADC TPP-9476, in lines 324-328:

A report has previously shown that the internalization of TPP-9476, an ADC targeting IL-3Rα, was 3.5-fold higher compared to isotype mAb [55]. Our data show that internalization is 8.6-fold higher for 7G3-Accum compared to isotype mAb and up to 9.4-fold higher for A14-Accum compared to A14 [14].

Furthermore, we have gone farther and added and highlighted a part comparing our different Accum-modified products with few peptides-modified products (“Discussion” section, Lines 350-367) to provide more insight:

Besides CPPs, other targeting peptides, such as p28, TCP-1, and LyP-1, rely on receptor recognition to deliver therapeutic payloads [72]. p28 is an amino acid that enables the copper-containing redox protein azurin to target cancer cells, eventually inducing cell cycle arrest [73]. In tumor-bearing subjects, it has been shown that p28 induces 6.6% CR, 20% PR, and 20% OS [74]. In tumor-bearing mice, we have shown that Accum-OVA vaccine maintains 30% OS alone and 100% OS in combination with adjuvants. Combined with anti-PD-1, Accum-DC vaccine also induced 10%–20% CR, 30%–40% PR, and 50% OS [15]. TCP-1, a peptide targeting the vasculature of gastric cancer cells, was conjugated to tumor necrosis factor α (TNFα) to deliver the anticancer agent 5-fluorouracil (5-FU). The results have shown that TCP-1 modification did not enhance the bioactivity of TNFα, as no differences in 5-FU-induced cancer cell death were detected [75]. Tumor volume also did not significantly change [75]. Accum-modification, however, was reported to enhance the bioactivity of T-DM1, markedly increasing its cytotoxic potency [10]. The tumor homing peptide LyP-1 was conjugated to abraxane, a clinically approved paclitaxel-albumin nanoparticle, then administered to mice bearing human cancer xenografts. Data have shown statistically significant results with 1.38-fold increase in cytotoxicity [76]. With Accum, however, the cytotoxicity of the clinically approved T-DM1 was enhanced by several folds [10].    

This manuscript is a resubmission of an earlier submission. The following is a list of the peer review reports and author responses from that submission.

Round 1

Reviewer 1 Report

Review of:

 AccumTM Technology: A Novel Conjugable Primer for Onco-Immunotherapy

This is a review about a cell targeting conjugate "Accum". The authors have developed multiple applications of this cancer cell targeting agent, which contains both a ChAc (Cholic acid) and an NLS (Nuclear Localization Signal) moiety. The authors review their own publications where they have conjugated Accum to monoclonal antibodies against IL3a, Il-5a. Furthermore, Accum was conjugated to T-DM1 (Trastuzumab emtansine), Ovalbumine and tumor lysate. Only the last example involves an immunotherapy application.

Each of these publications in their own right seem to show the advantage of conjugating Accum to cancer cell-targeting, cytotoxic and antigen-presenting moieties. 

Unfortunately, only Accum publications were reviewed. A more extensive scientific review including other competing technologies would be of much greater interest to the scientific community. The authors should have done that.

Reviewer 2 Report

Author described Accum as a novel biotechnology showing superior intracellular accumulation by escaping endosome and prompting nuclear targeting. From basic property of Accum to advanced studies including mechanistic in vitro stud and clinical study collaborated with immunotherapy, several interesting evidences to show the intelligence of Accum was clearly described. Following minor comments may be considerable for further improvement of the review manuscript.

The number of the subtitle of “3.1.7. G3-Accum” in “3. What has AccumTM…” seems to be strange.

As study using Accum is ongoing, detailed perspectives for clinical use of Accum should be included.

In the description regarding the G3-Accum in proof-of-concept experiments, significant and critical intracellular dynamics of G3-Accum modified monoclonal antibody are included, while there are no references that we could refer. All information is derived from Ref. 12? If so, more variety of references should be added to review the property of G3-Accum as a review article. Regarding the case of A14-Accum, T-DM1-Accum, OVA- and Lymphoma Lysate Proteins-Accum, available references seems to be limited. These descriptions may lead to biased concepts, which should be improved by adding a variety of references, sometimes comparison with other modified compounds may be effective.